

# TSECfire v1.0: Quantifying Wildfire Drivers and Predictability in Boreal Peatlands Using a Two-Step Error-Correcting Machine Learning Framework

Rongyun Tang[1], Mingzhou Jin[1], Jiafu Mao[2], Daniel Ricciuto[2], Anping Chen[3], Yulong Zhang[1]

[1]Institute for a Secure and Sustainable Environment and Department of Industrial and Systems Engineering, University of Tennessee, Knoxville, 37996, USA
[2]Environmental Sciences Division and Climate Change Science Institute, Oak Ridge National Laboratory, Oak Ridge, 37830, USA
[3]Department of Biology and Graduate Degree Program in Ecology, Colorado State University, Fort Collins, 80523, USA

*Correspondence to*: Mingzhou Jin(jin@utk.edu) and Jiafu Mao(maoj@ornl.gov)

**Abstract.** Wildfires are becoming an increasing challenge to the sustainability of boreal peatland (BP) ecosystems and can alter the stability of boreal carbon storage. However, a quantitative understanding of natural and anthropogenic influences on the changes in BP fires remains elusive. Here, we quantified the predictability of BP fires and their primary controlling factors from 1997 to 2016 using a two-step correcting machine learning (ML) framework that combines multiple ML classifiers, regression models, and an error-correcting technique. We found that (1) the adopted oversampling algorithm effectively addressed the unbalanced data and improved the recall rate by 26.88%–48.62% when using multiple datasets, and the error correcting technique tackled the overestimation of fire sizes during fire seasons, (2) non-parametric models outperformed parametric models in predicting fire occurrences, and the machine learning model of Random Forest performed the best with the area under the Receiver Operating Characteristic curve ranging from 0.83 to 0.93 across multiple fire data sets, and (3) four sets of factor-control simulations consistently indicated the dominant role of temperature, air dryness, and climate extreme (i.e., frost) for boreal peatland fires, overriding the effects of precipitation, wind speed, and human activities. Our findings demonstrate the efficiency and accuracy of ML techniques in BP fire prediction and disentangle the primary factors determining BP fires, which are critical for predicting future fire risks under climate change.

## 1 Introduction

The carbon-rich boreal peatlands (BPs) cover only ~2% of the Earth's surface (Gorham, 1991) but have accumulated ~20%–40% (450 ± 150 PgC) of the global soil carbon, historically playing a net cooling effect on the global radiation balance (Hugelius et al., 2020; Page and Hooijer, 2016; Scharlemann et al., 2014). This major land carbon pool, however, is highly vulnerable to current global warming, which tends to induce carbon emissions into the atmosphere through increasing decomposition of peat soil organic matter and fire combustions (Turetsky et al., 2014). In particular, BP fire regimes have





been undergoing pronounced changes over recent decades in terms of fire extent, frequency, and duration (Field and Raupach, 2004; Kelly et al., 2013). In BPs, there are two types of wildfires—surface flaming and underground smouldering—that can transition from one to the other at different phases. It is noteworthy that compared to flaming combustion, smouldering combustion is easier to ignite, harder to suppress, more persistent in low temperature and high

moisture peat (Huang and Rein, 2019). Besides releasing CO2, smouldering produces more CO, CH4, smokes, and even gaseous mercury (Haynes et al., 2017; Urbanski et al., 2008), altering global carbon balance and threatening public health (Liu et al., 2015; Reid et al., 2016). Yet smouldering combustion remains poorly understood, despite recent efforts on using experimental, statistical, and computational tools to investigate smouldering ignition, spread, extinction, fuel types, burning depth, and emission estimation (Che Azmi et al., 2021; French et al., 2004; Rein and Huang, 2021). As a consequence,

smouldering is not fully characterized in prevalent wildfire physical models (Rabin et al., 2017), although peatland fires are thought to be modulated by heat transfer and water content (Frandsen, 1997; Ohlemiller, 1985). Without an improved understanding of smouldering fires, therefore, our current understanding of BP fires and their predictability are still very limited, hampering the peat fire hazard mitigation and firefighting.

Most studies ascribe the ignition and propagation of flaming fires to the joint impact of heat source, fire-favor climate, fuel,

and anthropogenic factors. Flameless smouldering peatland fires are not an exception although upland flaming fires and underground smouldering in BPs are fundamentally different in their chemical and physical aspects (Certini, 2014; Costafreda-Aumedes et al., 2017; Rabin et al., 2017). However, compared to our understanding of flaming fires and their drivers and burning processes (Rothermel, 1972), we still know very little about key factors controlling smouldering fires. Importantly, Yuan et al. (2021) suggested that the smouldering process is a series of exothermic and often nonlinear events

that include three key steps: biological reaction, chemical oxidative reaction, and drying. However, quantifying the exothermic process is not easy. For example, experiments using phospholipid fatty acid (PLFA)-based microorganism revealed that peat self-heating reactions (soil respiration and microorganism growth) could happen at temperatures as low as 25–55 ℃ (Ranneklev and Bååth, 2003), while temperature could reach 500–700 ℃ during smouldering (Hurley et al., 2015). The dramatic changes in micro process of smouldering reactions consequently bring difficulties and uncertainties in

measuring parameters for physical models. Furthermore, without a clear understanding of nonlinear interactions of climate, heat transformation and fire, the use of traditional bottom-up statistic models can be clueless.

Rather than traditional linear models, more complicated process-based physical models and data-driven statistical models—including machine learning (ML) techniques—have been extensively used to explore the environmental determinants and predictability of peat wildfires (Bedia et al., 2014; Burgan and Rothermel, 1984; Castelli et al., 2015). Process-based fire

models are primarily based on well-established mathematical or physical laws that can describe fire processes, but these models may struggle with uncertain initiation and boundary conditions, and model parameters (Hantson et al., 2016). According to the Fire Modelling Intercomparison Project (Rabin et al., 2017), most fire schemes in current land surface models focus on forest fire occurrence, spread, distinction, and associated impact assessment. Only few models (e.g., the Community Land Model [Li et al., 2013; Rabin et al., 2017]) explicitly characterize peatland fire impacts with constrains



from climate (e.g., BP wetness and tropical dryness), peat fraction, water table depth, and grid cell area (Li et al., 2013). Substantial gaps in the knowledges and understanding of peat fire combustions, the solution of the primal and inverse problems, and the unavailable large scale peat soil and peat burning characteristic data are still obstacles in building the peat fire combustion theory and parameterizing peat fire in process-based models (Grishin et al., 2009). Unlike general statistic models which require assumptions and unlike physical models which are supported by physical mechanisms, ML models

require very few assumptions and can achieve high performance in solving nonlinear fitting and predictions (Jain et al., 2020). These benefits have stirred the application of a broad range of ML algorithms in wildfire science research, such as fire detection, fire weather exploration, fire behaviour prediction, fire impacts evaluation, and fire management (Jain et al., 2020). ML algorithms are not only used to attribute the primary causes of fires (Yu et al., 2020) but also applied to model evaluation and diagnosis (Forkel et al., 2019). However, the majority of ML research focuses on forest fires, and just a small

number of recent studies have used ML in the study of BP fires. For example, Rudiyanto et al. (2018) applied artificial intelligence in peatland monitoring and mapping with the support of remote sensing data, while some others investigated peat fire risk prediction and attribution with different ML methods (Bali et al., 2021; Horton et al., 2021; Rosadi et al., 2020). However, it is noteworthy that the recall or precision rate of peat fires was typically low in these ML studies, despite generally high (>70%) prediction accuracies (Bali et al., 2021; Horton et al., 2021; Rosadi et al., 2020). These low recall or

precision rates (i.e., high Type I and Type II errors) are likely caused by unbalanced fire data, which also indicated that predicting severely unbalanced fire by single models could be still full of challenges, and further studies are needed to deal with such commission and omission problems and to improve the predictability of peat fires.

For that reason, by collating and harmonizing monthly climate-, vegetation-, soil-, and human-related variables from 1997 to 2015, we created a two-step ML framework with various ML classification and regression techniques to evaluate the model

reproducibility and predictability on severely skewed fire data, and a series of sensitivity tests were performed on each of multiple fire data sets to address possible drivers of BP fires. Specific research goals include to (1) examine the performances of multiple ML algorithms on reproducing and predicting fire occurrence, fire counts, and fire impacts (i.e., burned area and carbon emissions), (2) diagnose dominant environmental controls on peatland fire activities, and (3) quantify uncertainties in the two-step ML framework and correct predicting errors to improve the ML predicting accuracy

that is suppressed by the severely skewed input data.

## 2 Data

Multiple sources of environmental data—including climate-, vegetation-, soil- and human-related data—and multiple fire products were used in this study, as listed in Table S1. All data sets were regridded to 1° × 1° with a monthly time resolution.



## 2.1 Response Variables

To evaluate ML framework robustness for difference response variables, five fire data sets were used in this study: the
Global Fire Emission Database (GFED) version 4.1s (GFED4.1s) carbon emissions, the GFED4.1s burned area (BA), the
Fire Climate Change Initiative (FireCCI) version 5.1 (FireCCI5.1) BA, the Moderate Resolution Imaging Spectroradiometer
(MODIS) active fire products MCD45A1, and MCD64A1 burning date. The monthly BA fraction and carbon emissions
from GFED4.1s span from 1997 to 2016 with a spatial resolution of $0.25° \times 0.25°$ (Giglio et al., 2013; Randerson et al.,
2012). The FireCCI5.1 BA data set ranges from 2001 to present and has a spatial resolution of 250 m at monthly or biweekly
temporal resolutions (Chuvieco et al., 2018; Lizundia-Loiola et al., 2020). Monthly MCD45A1 and MCD64A1 burn date
data sets were derived from the MODIS Terra and Aqua satellites products at a spatial resolution of 500 m. MCD45A1 was
derived from surface reflectance dynamics by a bidirectional reflectance distribution function–based change detection
approach (Roy et al., 2002), whereas MCD64A1 was produced by a burn-sensitive vegetation index algorithm based on a
combination of reflectance data and active fire observations (Giglio et al., 2018). Because only burn dates were provided,
both MCD45A1 and MCD64A1 were only applied for evaluating fire occurrences rather than fire impacts.

## 2.2 Explanatory Variables

### 2.2.1 Meteorology Data

To reflect the climate from 1997 to 2016, this study used the monthly $0.5° \times 0.5°$ gridded Climatic Research Unit (CRU)
Time-Series data version 4.04 (Harris et al., 2020). CRU data provide meteorological variables, including mean temperature
(TMP), temperature minimum (TMN), temperature maximum (TMX), cloud cover (CLD), diurnal temperature range (DTR),
ground frost frequency (FRS), wet day frequency (WET), evapotranspiration (ET), precipitation (PRE), and vapor pressure
(VP). Additionally, the CRU Palmer Drought Severity Index (PDSI) and the Modern-Era Retrospective analysis for
Research and Applications Version 2 (MERRA-2) 2m windspeed (WIN) were included as feature inputs. Using the CRU
saturated VP (SVP) and relative humidity (RH), we also calculated the VP deficit (VPD) based on the transforming
formulations shown in Table S1.

### 2.2.2 Vegetation Data

Monthly third-generation Global Inventory Monitoring and Modeling System (GIMMS-3g) NDVI from 1982 to 2015 with a
spatial resolution of $0.83° \times 0.83°$ was used to characterize the vegetation growth condition (Pinzon and Tucker, 2014). The
8 km gridded monthly GIMMS-3g gross primary productivity (GPP) from 1982 to 2016 were also included in this study to
characterize the fuel availability (Madani and Parazoo, 2020).



### 2.2.3 Soil Moisture Data

To estimate the effects of soil moisture on BP fire initiation and expansion, the Global Land Evaporation Amsterdam Model (version 3.3) surface soil moisture (SMsurf) and root-zone soil moisture (SMroot) were used (Martens et al., 2017; Miralles et al., 2011). These two datasets, which range from 1980 to 2018, were gridded at a spatial resolution of $0.5° \times 0.5°$ for each month.

### 2.2.4 Human Activity Data

The population density data were used as a proxy for human activities. The History Database of the Global Environment (version 3.2) were interpolated and re-gridded into a monthly scale at a spatial resolution of $0.5° \times 0.5°$ (Klein Goldewijk et al., 2017).

Multiple sources of environmental data—including climate-, vegetation-, soil- and human-related data—and multiple fire products were used in this study, as listed in Table S1. All data sets were regridded to $1° \times 1°$ with a monthly time resolution.

### 3 Methods

This study proposes a two-step error-correcting ML framework that includes the classification and regression steps. The classification step (Step One) primarily predicts fire occurrence and fire counts, whereas the regression step (Step Two) primarily predicts fire impacts if a peat fire occurs. The evaluation metrics from Step One, denoting the model uncertainties, are used at Step Two to correct fire size prediction uncertainties. The two-step ML framework is detailed in Figure 1.

Five fire datasets were applied as the target variable separately at Step One: FireCCI5 BA, GFED BA, GFED carbon, MCD45A1, and MCD64A1 active fires, and three of these datasets—FireCCI BA, GFED BA, and GFED carbon—were also used in Step Two. Because of the notable imbalances of fire occurrence data (i.e., there are more nonoccurrence records than occurrence records), multiple principal evaluation metrics were checked to evaluate ML performance in predicting fire occurrences at Step One. For each dataset and simulation, all evaluation metrics were extracted and ensembled from all six ML algorithms. Evaluation accuracy results are listed in Tables S2–S5.



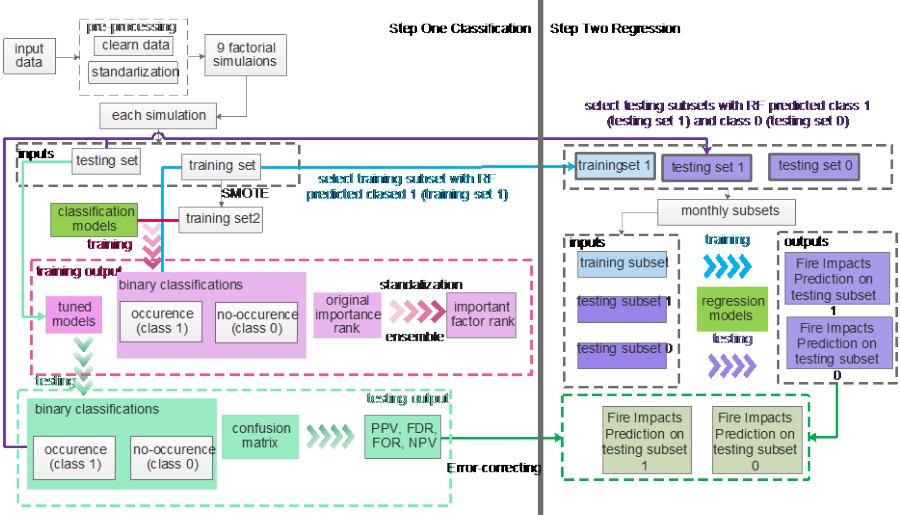

**Figure 1. The two-step ML framework, where PPV, FDR, FOR, and NPV stand for positive predictive value, false discovery rate, false omission rate, and negative predictive value, respectively. SMOTE stands for the oversampling algorithm– Synthetic Minority Oversampling Techniques. The error-correcting process is detailed in the Methods part.**

We first preprocessed the data, including data integration, missing values treatment, and standardization, and then randomly split the data by 70% for training and 30% for testing. An oversampling algorithm called Synthetic Minority Oversampling Techniques (SMOTE) was applied onto the training dataset to address the imbalance between the two fire occurrence classes.

In Step One, six common classification algorithms—logistic regression (LogR), linear support vector machines (SVMs), Random Forest (RF), Bagging (BAG), k-nearest neighbors (KNN), and Gaussian Naïve Bayes (GNB)—were applied to classify fire occurrence at each grid in each month. Then, key factors driving peat fire occurrences were ranked by these ML algorithms to find the feature subset with large contributions. Feature importance values in RF and BAG were calculated as the mean decrease in node impurity (i.e., Gini index) weighted by reaching probability of samples to each node. For LogR and SVM, the coefficients of the features in LogR's decision functions and in linear SVM's weights were extracted to assess feature importance. The KNN and GNB classifiers did not offer direct ways to assess feature importance. This study used a permutation method to assess feature importance based on the loss function and increased prediction error after shuffling features. Because the feature importance values assessed from different ways were not in the same value range, feature importance values were processed using the normalized absolute value for a consistent comparison. The mean and standard variation of the normalized feature importance values from different ML models characterized the relative importance of driving factors and model differences.

Using the predicted monthly fire occurrence from the best-performing ML classifier, the corresponding fire data were retrieved to predict fire impacts (including burned area and C emissions). For months without fire occurrences, the fire impacts were estimated as zero before error correction. In Step Two, 14 regression techniques—simple linear, ridge, least



absolute shrinkage and selection operator (LASSO), adaptive boosting (AdaBoost), gradient boosting, Bagging, RF, Bayesian, elastic net, kernel ridge, decision tree, CatBoost, and light gradient boosting—were tested to predict fire impacts. The study further corrected the predicted fire impacts by applying evaluation metrics of classifications—including the

positive predictive value (PPV, namely precision), false discovery rate (FDR), false omission rate (FOR), and negative predictive value (NPV)—to amend the classification uncertainties in the regression. The details of the error-correction are below.

After classification, in the training dataset, C emissions/Burned area equals 0 are classified as no-fire months, which is denoted as class 0, while values greater than 0 are classified as fire months which are correspondingly denoted as class 1. In

the original training set, samples are separated into fire months ($X_{mf}$) and months with no fire ($X_{mn}$).

Here, we selected samples with C emissions/Burned area greater than 0 (namely class 1) to train the fire regression model for the month m, $R_{mf}$:

$$R_{mf}(X_{mf}) = Y_{mf}, \tag{1}$$

Where $X_{mf}$ is the explanatory data in fire month $m$; and $Y_{mf}$ is predictive variable (C emission/burned area) at month $m$.

For month m with no fires, we suppose regression model $R_{mn}$:

$$R_{mn}(X_{mn}) = 0, \tag{2}$$

where $X_{mn}$ is the explanatory data in month m with no fires.

For each month ($m$), we have split the training dataset into $X_{mf} (with\ fires)$ and $X_{mn} (without\ fires)$, and split the testing dataset into $X'_{fm} (with\ fires)$ and $X'_{nm}(without\ fires)$. Keeping the input data as the same, 14 regressors are applied in

this experiment, and they are: Linear Regressor, Ridge, Lasso, AdaBoost Regressor, Gradient Boosting Regressor, Bagging Regressor, Random Forest Regressor, Bayesian Ridge Regressor, Elastic Net Regressor, Kernel Ridge Regressor, Decision Tree Regressor, CatBoost Regressor, LGBM Regressor, and Stacking Regressor.

For each month ($m$) in {1,2,3 … 12} and regression model ($R^r$) in {$R^1$, $R^2, R^3 … R^{15}$}, we constructed regression models $R^r_{mf}$ for fires at month m and $R^{r,m}_n$ for month m without fires:

$$R^r_{mf}(X_{mf}) = y^r_{mf}, \tag{3}$$

$$R^r_{mn}(X_{mn}) = y^r_{nm} = 0, \tag{4}$$

Then, with testing data, we do prediction of fire size using the above-trained regression model $R^r_{mf}$.

For fire month m (class 1) in testing data, the predicted fire size $P^r_{mf}$:

$$P^r_{mf} = R^r_{mf}(X'_{mf}), \tag{5}$$

Fire size that might be caused by the wrong classification (namely, no fire happens in reality) could be expressed by $EP_{nm}$:

$$EP^r_{mn} = R^r_{mn}(X'_{mf}) = 0, \tag{6}$$

While for months without fires (class 0), the predicted fire size $P^r_{mn}$:

$$P^r_{mn} = R^r_{mn}(X'_{mn}) = 0, \tag{7}$$





And the fire size that might be caused by the wrong classification (namely, fire happens in reality) could be expressed by:

$$EP^r_{mf} = R^r_{mf}(X'_{mn}),\tag{8}$$

Four evaluation metrics from classification are used to adjust prediction uncertainties, and they are:

$$Positive\ predictive\ value(PPV) = \frac{True\ Positive\ (TP)}{TP+False\ positive(FP)},\tag{9}$$

$$False\ Discovery\ rate\ (FDR) = \frac{FP}{TP+FP},\tag{10}$$

$$False\ omission\ rate\ (FOR) = \frac{False\ Negative\ (FN)}{FN+True\ nagative(TN)},\tag{11}$$

$$Negative\ predictive\ value\ (NPV) = \frac{TN}{FN+TN},\tag{12}$$

Applying classification evaluation metrics to the actual predictions (P) and potential wrong classification caused predictions (EPs), we could obtain the error-corrected prediction $AP'^r_{mp}$ for the record of $(p,m) \in X'$ (in the testing set).

$$AP'^r_{mp} = \begin{cases} PPV \times P^r_{mf} + FDR \times EP^r_{mn}, & If\ Z'_{mp} = 1 \\ NPV \times P^r_{mn} + FOR \times EP^r_{mf}, & If\ Z'_{mp} = 0 \end{cases}\tag{13}$$

Where $(f,n) \in p, and\ Z'_{mp}$ stands for the original prediction in testing data.

To validate the ranking of feature importance from MLs, a range of factorial simulations that grouped features with similar physical meanings were designed and conducted. The temperature-related group contains TMP, TMN, and TMX; PRE is the only PRE-related feature; the air dryness-related group include SVP, VAP, VPD, RH, WET, ET, and PDSI; the soil moisture–related features are SMsurf and SMroot; and the Others group includes features representing vegetation biomass (e.g., GPP and NDVI), windspeed (WIN), cloud cover percentage (CLD), climate extremes (e.g., FRS and DTR), and

anthropogenic activities (e.g., POPD). During the first set simulation, we conducted the simulation ALL with all features. As will be shown in Section 4.3, the features in the Others group are generally ranked at low level. As the Other group contain various features with different physical meanings and the large variety of features could be one primary source of feature collinearity, the Others group features were also kept in the following three sets of simulations and we targeted to compare the other four groups of features (temperature group, precipitation group, air-dryness group, and soil moisture group). During

the second set of simulations, each simulation opted out one group of features to explore the top-ranked group among the four groups. For example, the TMP group features were removed in the NO-TMP simulation to compare the relative importance of the remaining feature groups (i.e., the PRE group, the air dryness group, the soil moisture group, the Others groups). Similarly, we excluded the PRE group, the air dryness group, the soil moisture group in the NO-PRE, NO-HUMI, and NO-SOM, respectively. The temperature group was consistently ranked the highest in the NO-PRE, NO-HUMI, and

NO-SMO. During the third set of simulations, only the relative importance of the PRE group, air dryness group, soil moisture group, and the Others group needed to be compared because the TMP group was already identified as the top-ranked features. Thus, the TMP group was removed in all simulations in the third set. Additionally, The NO-TMP-PRE, NO-TMP-SOM, and NO-TMP-HUMI simulations were designed by respectively removing the PRE group, soil moisture group, and air dryness group to diagnose the relative importance of soil moisture group V.S. air dryness group, PRE group V.S. air





dryness group, and PRE group V.S. soil moisture group. The air dryness group ranked the highest in the third set of simulations, and PRE ranked the lowest in the first, second, and third sets of simulations. Based on this, we continued setting up the fourth set of simulations, NO-TMP-PRE-HUMI, to check the relative importance between soil moisture and the other vegetation and human factors. Table 1 lists the experimental designs of simulations.

**Table 1. Simulation experiments for assessing environmental factor cluster impacts on ML predictability**

| Simulations | | Explanatory variable groups | | | | |
|---|---|---|---|---|---|---|
| | | Temperature-related | Precipitation-related | Air-dryness related (i.e., Humidity) | Soil moisture-related | Others |
| First | ALL | Yes | Yes | Yes | Yes | Yes |
| Second | NO-TMP | No | Yes | Yes | Yes | Yes |
| | NO-PRE | Yes | No | Yes | Yes | Yes |
| | NO-HUMI | Yes | Yes | No | Yes | Yes |
| | NO-SOM | Yes | Yes | Yes | No | Yes |
| Third | NO-TMP-PRE | No | No | Yes | Yes | Yes |
| | NO-TMP-SOM | No | Yes | Yes | No | Yes |
| | NO-TMP-HUMI | No | Yes | No | Yes | Yes |
| Fourth | NO-TMP-PRE-HUMI | No | No | No | Yes | Yes |

**4 Results**

**4.1 Fire Occurrence Predictability**

The averaged area under the receiver operating characteristic curve (AUC) which indicate the diagnostic ability of classification ranged from $0.70 \pm 0.03$ (MCD64A1, the No-TMP-PRE-HUMI simulation) to $0.88 \pm 0.05$ (MCD45A1, the ALL simulation) for multiple MLTs (Table S2). The ALL simulation had the AUC value of 1 at the training stage and the

AUC value of 0.72–0.93 at the testing stage. The RF algorithm showed the best predictive performance for fire occurrences (i.e., fire counts) (Table S3) and provided a basis for fire impacts prediction. Among all datasets, MCD45A1 had the highest recall rate (0.94) and highest precision (0.96), indicating that few months were incorrectly classified (Table S3). MCD64A1 had the lowest recall rate and precision rate, indicating discrepancies among different data sources. Using the SMOTE oversampling algorithm, the testing recall rate was effectively improved at an average rate of 26.88% and with the highest

growth of 48.62% for the FireCCI BA dataset (Tables S4 and S5).

Besides evaluation metrics, the spatial disparities of predicted fires from MLTs and multiple datasets were also examined against corresponding observations. The BP with a histosol fraction greater than 30% is mainly located in the Hudson Bay Lowland (HBL) and West Siberia (WS) (Figure S2). Observations from FireCCI BA, GFED BA, GFED carbon emissions, and MCD64A1 fire detection consistently show that there were fewer than 60 fire events in the HBL region from 1997 to

2015, but the fire count in WS during the same time period ranged from 30 to more than 150. This demonstrates the spatial disparity of peatland fire occurrences in the boreal area and possibly implies that WS is more fire-prone than the HBL (Figures S3–S6[a-1], [a-2]). FireCCI, GFED, and MCD64A1 showed good consistencies among these three products with





respect to the data distribution. Unlike these three datasets, MCD45A1 had higher estimation and lower spatial heterogeneity of fire counts in BP (Figures S7[a-1], [a-2]). The more evenly distributed data in MCD45A1 may be the primary reason why

MCD45A1 had the highest predicting accuracy and best performance in reproducing the distribution of fire counts spatially (Figure S12) and temporally (Figure S16) in the testing stage (Figures S8–S11 and S13–S15).

Predictability discrepancies were also compared among multiple ML algorithms. The validation results demonstrate that the bootstrap-based ML algorithms (i.e., RF, BAG, and KNN)—in which there is no requirement for data distribution assumption, and resampling supports the inference of the population distribution—had better predictability than other

algorithms (i.e., LogR, linear SVM, and GNB) (Figure S1 and Figure 2). For RF and BAG, the reproducing accuracy rate (i.e., true positive rate and true negative rate) was higher than 90% for the FireCCI data (Figure S1). The inaccurate predictions of KNN, LogReg, SVM, and GNB were significantly influenced by the overestimated fire occurrence (namely false positive) during fire season (April–October), as shown in Figure 2. Without a prescribed underlying function, the nonparametric RF and BAG models exhibited advantages over other ML algorithms on reproducing peatland fire

distributions spatially (Figures S9–S12[b-1], [b-2], [c-1], [c-2], [d-1], [d-2]) and temporally (Figure 2, Figures S13–S16[a-1], [b-1]). Therefore, the predictions of fire occurrence from the best-performed RF were employed as the basis of fire impact predictions.

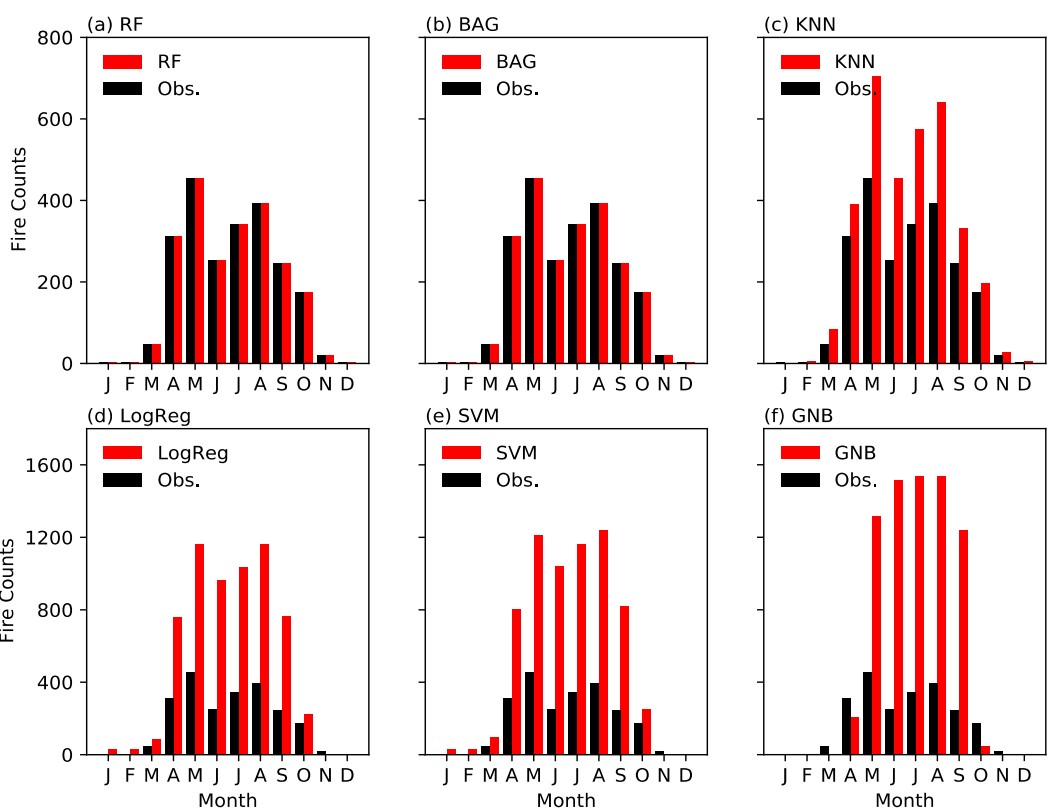

**Figure 2. Seasonality of observational and predicted fire counts from the six ML algorithms with the FireCCI BA dataset.**



## 4.2 Fire Impacts Predictability

ML regression models exhibit moderate predictabilities of fire sizes (Figure S35). ML classification at Step One and regression models at Step Two overestimated fire size during fire season (Figures S29–S31) for the monthly aggregated fire impacts. This study developed an error-correcting technique to tackle such overestimation during fire reason and achieved satisfying performance (Figure 3, Figures S29–S31).

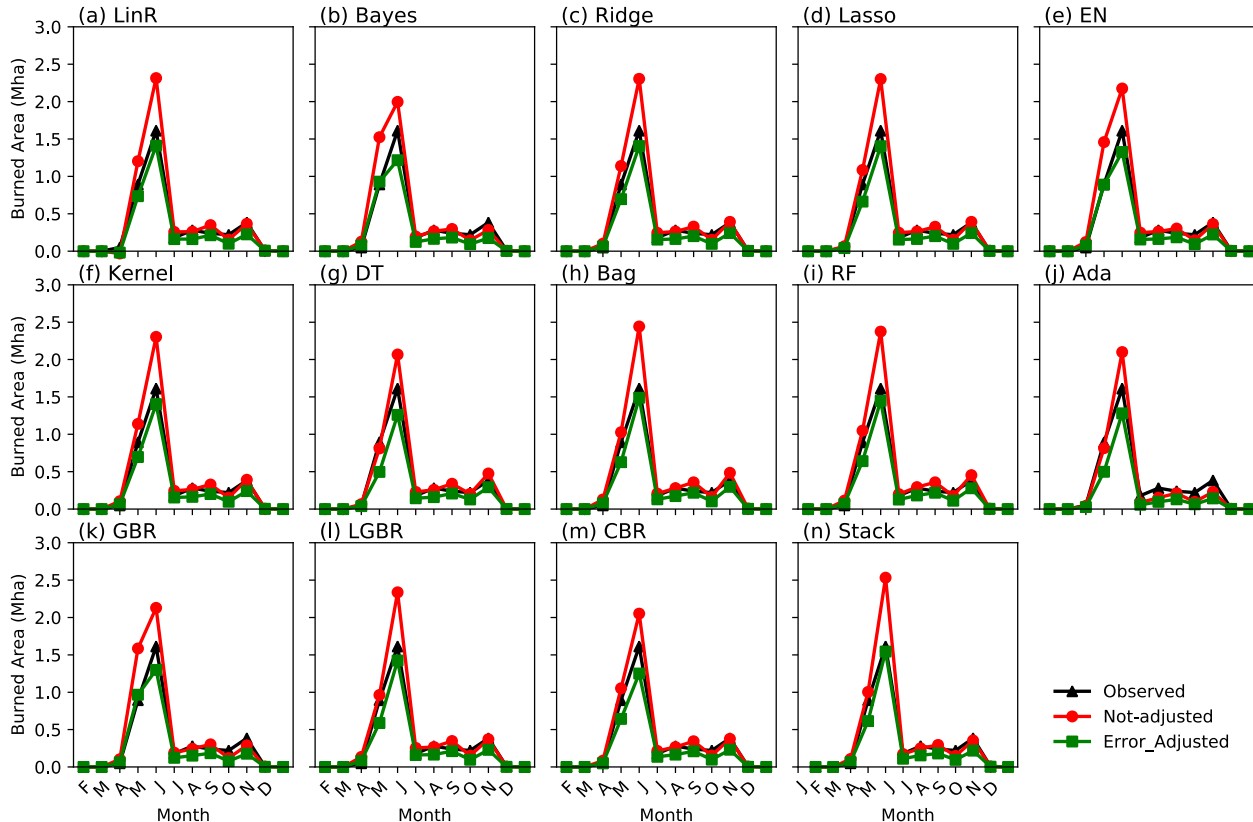

**Figure 3. Seasonality of the observed, not-adjusted, and error-adjusted FireCCI BA based on the testing phase from multiple ML regression models: (a) linear, (b) Bayesian linear, (c) ridge, (d) LASSO, (e) elastic net, (f) kernel ridge, (g) decision tree, (h) Bagging, (i) RF, (j) AdaBoost, (k) gradient boosting, (l) light gradient boosting, (m) CatBoost, and (n) stacking.**

WS has more fire counts and thus higher carbon emissions than the HBL (Figures S26–28). The predicted carbon emissions from the stacked ML algorithms were overall consistent with the observations in WS and western Canada but had overestimations in the HBL (Figure S28). The error-correcting technique could slightly lower the overestimation in the HBL (Figure S28) but could greatly lower the overestimation temporally, especially in July (Figure S31). Meanwhile, the underestimation of fire impacts in June remained a common problem for all 14 regression models (Figure S31).

GFED BA and FireCCI BA were used to determine the reliability of fire impact predictions within the two-step ML framework. In terms of spatial reproducibility, the predictions from GFED BA (Figures S30[d], [e], [f]) were more accurate than those from FireCCI BA (Figures S29[a], [b], [c]), particularly in the HBL, where the BA is less than 50 km$^2$ (Figures



S26–S28[a-1], [b-1]). Figures S26–S28 [a-2] and [b-2] show that the framework underestimated burned area in northern WS and overestimated burned area in the northern HBL for FireCCI BA (Figures S26–S28[a-1], [b-1]). Different BA datasets can also have temporal inconsistencies. FireCCI BA exhibits its fire season from March to May, whereas GFED BA exhibits

its fire season from March to October. Despite the fact that April and May were the fire peak months according to both FireCCI BA and GFED BA, the burned areas predicted by the framework based on FireCCI and GFED show differences in BP. According to FireCCI, the predicted entire burned area in May has about 55,792 km$^2$, whereas the prediction based on GFED is only about 12,183 km$^2$. A further investigation shows that GFED BA has a bimodal distribution while FireCCI BA is unimodally distributed (Figure S30). Therefore, it is important to determine whether ML is applicable for various datasets.

Overall, the 14 tested regression models were able to well reproduce fire impact magnitudes and seasonality for the FireCCI BA, GFED BA, and GFED carbon (Figures S32–34). Figures 3, S30, and S31 show that for all datasets, the ML regression models appear to overestimate the fire effects, including carbon emissions and burned area during fire season. However, the error-correcting approach could successfully reduce this bias (Figures S29–34). Discrepancies among model predictabilities were small. For example, for the FireCCI data, the decision tree had the best performance with estimations that were 4.05%

higher than the observations, whereas Bagging had the worst performance with estimations that were 10.84% higher than the observations. Such small biases and discrepancies verified the reproducibility and predictability of the two-step ML framework.

## 4.3 Primary Causes of BP Fires

To exclude feature collinearity, four sets of simulations in Table 1 were designed by opting out grouped features to confirm

the importance ranking of features (Figure 4). The first two sets of simulations (i.e., ALL, NO-TEMP, NO-PRE, NO-HUMI, and NO-SOIMOI) showed that the temperature-related feature group had the highest importance (Figure s3[a], [c], [d], [e]). The third sect of simulations, which removed two feature groups, showed that the air dryness had the highest importance among the remaining four feature groups, namely the PRE, air dryness, soil dryness, and other groups. PRE was found to be the third-ranked feature according to the first three sets of simulations. The last set simulation was conducted to compare the

relative importance of soil moisture and the other human and natural features and found that frost (FRS) and vegetation biomass (GPP) in the other human and natural features group were more important than soil moisture (Figure 3[i]). Such ranks were also indicated by other simulations in Figures 3[a], [c], [d], [f], and [h]. Thus, this study found that BP fires were significantly affected by temperature, air dryness, frost, and GPP (Figure 3[a]), which collectively account for more than 80% of the predictive interpretability (Figure 3[a]). Moreover, BP fires were not sensitive to PRE, soil moisture, windspeed,

and human activities.





**Figure 4. The bar plot stands for the factor importance rank of multiple simulation scenarios using FireCCI BA as the target variable in which the importance was determined by standardized mean and uncertainty range (minimum and maximum) from multiple ML algorithms; the dashed vertical line indicates the group mean importance of temperature (blue), PRE (yellow), air dryness (purple), soil moisture (orange), and other factors (green).**

The feature importance ranks were not only validated by FireCCI BA but also by GFED BA, GFED carbon, MCD45A1, and MCD64A1. The rankings from GFED BA and GFED carbon were highly consistent with those from FireCCI (Figures 4, S22, and S23), in which temperature, air dryness, frost, and GPP were more important than PRE, soil moisture, windspeed, and other natural and anthropogenic factors. Feature rank discrepancies were found when the ML algorithms were applied to



MCD64A1 and MCD45A1, for which the top three features were still air dryness, temperature, and FRS, but soil moisture was more significant than GPP (Figures S24 and S25).

Collectively, the multisource datasets and multi-feature simulation experiments consistently suggested that air dryness–related variables (RH, VPD, and VAP), temperature-related variables (TMN, TMP, TMX) and FRS play more important roles in the peat fires than other factors, such as PRE, wind speed, and other natural and human factors. The importance of
soil moisture and GPP were both ranked in the middle, but their relative rankings could not be determined because soil moisture was considered more significant than GPP according to MCD64A1 and MCD45A1 but GPP was viewed more important based on FireCCI, GFED BA, and GFED carbon.

## 5. Discussion

### 5.1 ML Predictability

The global peatland contains ~25% of global soil carbon (600 GtC) (Yu et al., 2010) and is at risk of shifting from the world's largest carbon sink to the largest carbon source with warming climate and increasing fire events (Hugelius et al., 2020; Loisel et al., 2021; Turetsky et al., 2014). Predicting fire risks and fire impacts is extremely challenging given the inadequate representation of the peatland ecosystems and fire interruptions in current process-based models. As an alternative, ML algorithms can capture nonlinear relationships between the controlling factors and fire impacts (including
burned area and C emissions) and provide a unique method to explore fire driving mechanisms and predictability based on big data.

In this study, a two-step error-correcting framework was built to investigate the BP fire predictability and the individual impacts from meteorological, vegetational, soil, and anthropogenic factors. Although ML algorithms have been extensively used in the wildfire research (e.g., fire spots detection, predictive models development) (Coffield et al., 2019; Jain et al.,
2020; Sayad et al., 2019; Wang and Wang, 2020; Yu et al., 2020), few studies explicitly describe what the criteria they would use to choose the ML models. Predicting accuracy may depend on the modeling algorithms and the input data. In this study, results from six classification models and 14 regression models indicate that nonparametric ML algorithms, including RF, Bagging, and KNN, outperformed the other employed parametric models, such as LogR, linear SVM, and GNB, by overcoming the severe imbalance of fire data (the non-fire classes have six times as many records as fire classes) (Figures 1
and 2). Unlike parametric models that are highly restricted to specified functional forms and a fixed number of parameters, nonparametric models can fit various functional forms, and the number of parameters grow with the size of the training set, promoting the performance of model predictability.

In BPs, it is challenging to predict fire occurrence because of the extremely unbalanced fire data. Several previous studies have employed ML to investigate the peatland fire predictability. For example, Rosadi et al. (2020) employed a variety of
ML algorithms to predict fire occurrence in peatland and used the accuracy as the only evaluation metric. Such an evaluation method could fail to measure fire predictability once the fire data are imbalanced. According to another study that predicted





peatland fire occurrence in Canada (Bali et al., 2021), the recall rates were very high (0.82–0.99) but the precision metrics were very low (0.002–0.05), which indicates a high Type I error. In our study, RF regressions yielded high precision metrics (0.56–0.96) and recall rate (0.6–0.94) and well-identified fire months, suggesting relatively low Type I and Type II errors.

To address the extreme data imbalance, this study used both preprocessing (oversampling) and postprocessing (error correcting) in the two-step ML framework to improve the predictability. In Step One, the SMOTE algorithm significantly improved the recall rate by ~26.88%–48.66% across all fire datasets. Processing approaches (e.g., oversampling and undersampling) were also found beneficial in earlier studies for certain ML algorithms (Farquad and Bose, 2012; Malik et al., 2021; Zhou et al., 2020). To quantify and reduce uncertainty in ML frameworks, procedures are typically highly tailored

for specific research challenges and ML algorithms (Jiang and Nachum, 2020; Pan et al., 2019; Wang et al., 2020). In our two-step ML framework, applying evaluation metrics from the classification step (Step One) in error correcting effectively lowered the overestimated BA and carbon emissions during fire season (Figures S29–S31).

## 5.2 Primary Driving Factors of Peatland Fires

ML-derived statistical correlations do not necessarily indicate the causality, and biophysical or biochemical principles are

thus needed to further examine whether such relationships are reasonable (Schölkopf et al., 2021). In this study, four sets of ML simulations were designed to determine the primary driving factors of peatland fires by removing feature groups sequentially. The results revealed that the feature importance rank exhibited general consistency in multiple fire datasets. PRE in boreal or sub-arctic regions is primarily in form of snow rather than rainfall due to cold weather (Behrangi et al., 2016) and has little impact on BP fires. Moreover, smouldering fires can persist for a long time (months to years) even in

rainfall weather (Lin et al., 2020). This low importance was verified by our ML simulations. Similarly, in the sparsely populated boreal peatland, human activities showed marginal effect. Factorial simulations consistently demonstrated that temperature (i.e., minimum, maximum, and average values), air dryness–related variables (e.g., RH, VPD, VAP, ET), and FRS were the primary factors driving the BP wildfire activities (Figure 4 and Figure S22-23). Although these factors eventually lead to dry and combustible conditions for peatland fire occurrence and propagation, the processes in which they

play roles are quite different.


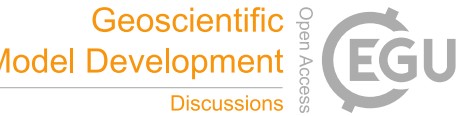


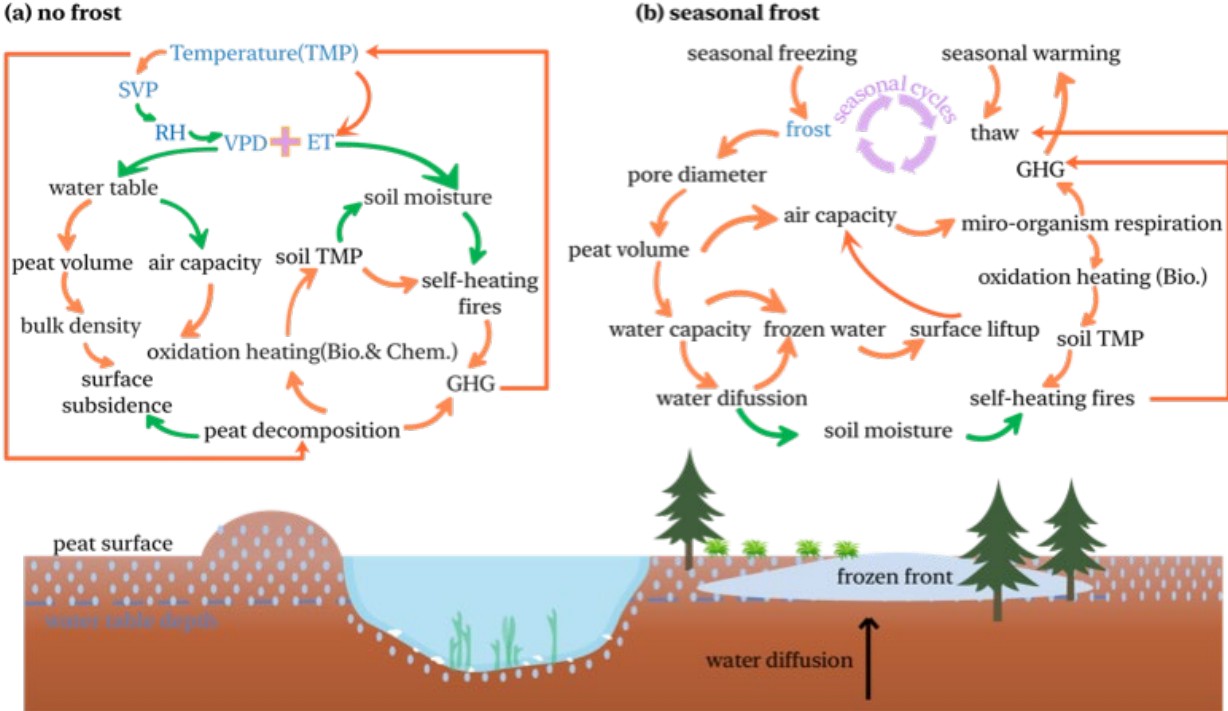

**Figure 5 Processes in which environmental factors participate for self-heating peatland fires. ML-identified primary factors are marked in blue; green arrows indicate negative correlation between the connected two factors, and orange arrows indicate positive correlation between the connected two factors.**

The BP fires are intimately tied to weather, and warming appears to increase ignitions, fire frequency, and fire severity (Duffy et al., 2005; Flannigan et al., 2005; Kohlenberg et al., 2018). In peatlands without frost (Figure 5[a]), rising temperatures increase saturation vapor pressure (SVP) and continually induce an increase in vapor pressure deficit (VPD) if actual atmospheric VP does not increase as much as SVP. A recent investigation indicates that RH (i.e., ratio of actual water VP to SVP) has plunged rapidly since year 2000, leading to a sharp rise in VPD on a global scale (Yuan et al., 2019). Such a

warming-induced increase of VPD increases evapotranspiration (ET) more in peatlands than in forests with a simulated percentage of up to 30% (Helbig et al., 2020). Because atmospheric demand (i.e., VPD) dominates the limitation of ET over the soil moisture (Helbig et al., 2020; Novick et al., 2016), the water table turns out to be the water supplier in response to the rising VPD, which consequently results in the decrease of water table depth. The water table depth decrease tends to change the physical characteristics of peat in many aspects, such as by lowering the capacity of water storage, causing the

peat volume to shrink and volumetric soil moisture to decrease (Price and Schlotzhauer, 1999), and inducing surface subsidence with a concomitant decrease of bulk density and an increase of peat oxidations and decomposition (Leifeld et al., 2011; Whittington and Price, 2006). These changes ultimately lead to more carbon being released into the atmosphere and the formation of dryer and more flammable peat soil (Figure 5[a]). In peatland with frost, frost heaving deepens the active



layers (Jones et al., 2015; Wang et al., 2020), changes the hydrological and thermal properties of peatland, promotes
microbial and chemical exothermic reactions, strengthens peatland dryness, and consequently facilitates more frequent
peatland fires (Kim et al., 2020) (Figure 5[b]).

Our ML-based sensitivity simulations demonstrated the power of using big data to determine the primary causes of peat fires: temperature, atmospheric dryness (e.g., RH, VAP, VPD, ET), and frosts (i.e., FRS). These simulations also helped identify the less important factors and processes. For example, wind speed and population density were ranked at the bottom,
suggesting that human activities may not be the main causes of peatland fire occurrence, and that the wind speed, unlike for forest fires, does not significantly affect peatland fire spread. Another intriguing discovery is that the simulations in this study consistently revealed the important role that FRS has played in causing peatland fires and their spreads, though FRS has been understudied in previous studies. Dixon et al. (2018) revealed that seasonal frost layer alters Spring water balance, induces drier Spring, and enhances risks of deep smouldering. More specifically, ground freezing frost can greatly change
the structure and properties of peatland. During the water icing process, the pore diameter is enlarged, which consequently results in peat volume expansion, water tension decreases, water storage capacity increase, and air capacity surges (Dijk and Boekel, 1965). As the air capacity increases, the oxidation of the soil organic carbon is likely increasingly. This oxidation produces heat and makes the soil temperature increase, which can start peatland fires by self-ignition (Arief et al., 2019; Restuccia et al., 2017). During the seasonal freezing process, soil water diffuses vertically from the bottom unfrozen layer to
the upper frozen layer (frost front) (Nagare et al., 2012). After cycles of freezing and thawing (i.e., frost heaving), surface peat soil becomes dryer, and the freezing surface becomes thicker in the form of surface lift above the water table. At low temperatures, heat generated from respiration and the growth of micro-organisms dominate heat generated from chemical oxidation in peat decomposition (Yuan et al., 2021). If frost heaving causes the peatland to dry out year by year, exothermic processes from biological reactions may intensify chemical oxidation with high temperature and thus induce spontaneous
peatland fires (Figure 5[b]).

Collectively, the important factors uncovered by the ML framework indicated two peatland fire mechanisms that suit two types of peat soil: unfrozen and seasonal frozen peatlands (Figures 5[a] and 5[b]). Temperature, air dryness, and the facilitated warming and drying in an underground environment may start fires in unfrozen peatland. For seasonal freezing and thawing in seasonal frozen peatland, frost heaving induces a deep drying and oxygen-rich underground environment and
may speed up exothermic progress in biological reactions, thereby promoting peatland fire occurrences.

There are several limitations of this study. Because of a lack of gridded burned depth data and bulk density, this ML-based work could not predict and evaluate peat fire severity. The satellite-based fire datasets used in this study do not provide underground smouldering peat fire as a single product. Fires detected by satellites could be a mixture of peatland surface flaming fires and smouldering fires because the detected radiant signature of smouldering is much weaker than that of
flaming fires (Rein and Huang, 2021). In addition, for peat fire C emissions, it has been estimated largely by multiplying detected burned area by a range of parameters, such as average burning depth, combustion completeness, emission factors of major carbon species. Those estimated parameters may induce large uncertainties due to the limited ability of optical





satellites to detect underground smouldering and burning depth (Graham et al., 2022). The limited data availability, such as vegetation types (moss and vascular plants), burning depth, bulk density, water table depth, and soil temperature, makes ML
algorithms limited in fully accounting for all contributing factors. Moreover, since the relationships identified by the ML framework do not automatically imply causality, the underlying physical mechanisms still need to be further validated by future experimental work or theoretical analyses, such as the overriding control of temperature-related variables on inducing boreal peatland fires and the mechanism by which frost impacts on peat drying and smouldering (Dixon et al., 2018).

## 6. Conclusion

This study constructed a two-step error-correcting ML framework to explore the predictability of peatland fire occurrences and impacts (including burned area and C emissions). Major climate, vegetation, soil, and human factors that possibly induce BP fires were included in a range of factorial simulations. The framework successfully predicted the fire counts (occurrences) and fire impacts with an accuracy in general greater than 80%. Temperature and air dryness were identified to dominate the fires in unfrozen BPs, while FRS was determined to dominate fire in frozen BPs through the impacts of frost
heaving (seasonal freezing-thawing) on changing thermal-hydrological characteristics of peat soil. Our research provides preliminary insights into the overriding impacts of temperature (including temperature related air-dryness and frost heaving) on BP fires via big data and ML. To overcome the ML's limitations in inferencing causality from data association and to further validate the underlying physical mechanisms in BPs fire, more field data (such as peat soil properties and peat burning properties) as well as additional experimental, statistical, or computational works are needed in the future.

*Code and data availability*: The model code and data that support the findings of this study are available at https://github.com/tangryun/GMD and data sources are also listed in the Supplementary Information.

*Autor Contributions:* MJ, JM, and RT conceived the research ideas. RT wrote the initial draft of the manuscript and performed the modelling and analysis. All authors contributed to preparation of the paper, edit the paper, draft revision, and provide scientific suggestions.

*Competing interests:* The authors declare that they have no conflict of interest.

*Acknowledgements.* This work was funded and supported by U.S Department of Energy. The authors thank the High Performance & Scientific Computing group of the University of Tennessee and Oak Ridge National Lab for providing the computing resources.



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
