# Peer review of "TSECfire v1.0: Quantifying Wildfire Drivers and Predictability in Boreal Peatlands Using a Two-Step Error-Correcting Machine Learning Framework"

_Geoscientific Model Development, 2023_

## Author Response (AR1)

**Reviewer 1:**

The authors present a two-step error-correcting machine learning framework, TSECfire v1.0, which is used to predict BP fires and their primary controlling factors from 1997 to 2016 in Boreal Peatlands. While the method has potential value, it is important to compare it to existing research, including other machine learning methods and data-driven fire models that have been developed in recent years. I recommend that the authors provide a more extensive evaluation of their method's accuracy and its unique contributions to the field in comparison to other existing methods. Additionally, the methodology section contains ambiguous expressions that require further clarification and specificity.

**Response to reviewer#1:**

Thank you for your valuable comments and suggestions. we have made update to the methodology section, with the changes highlighted in red. For more specific questions, we will address each question below.

1. The title of the study focuses on BP fires, but the paper does not provide sufficient clarification on how the method used in this study differs from other approaches used to predict BP fires. Please provide additional details to highlight the unique aspects of their methodology. In the Introduction section, there is a significant discussion on smouldering fires from BPs; but there is no mention of this phenomenon in the Method and Results sections. Can the GFED or FireCCI products distinguish between peatland fires that are smouldering or flaming?

   **Response:**

   We appreciate your insightful questions and suggestions on the need for additional clarification regarding our methodology and the phenomena of smoldering fires. We apologize for any lack of clarity in the initial draft.

   Regarding your first query, the methodology used in our study to predict boreal peatland (BP) fires presents several unique elements. Unlike wildfires in fire-prone regions (e.g., African savanna and the western United States), the BP fires often represent extreme events, which means that the fire data is of low frequency but large variance. The direct prediction of these extreme BP fires in terms of their fire sizes poses a significant challenge for two principal reasons: the paucity of available BP fire data to train the data-driven model (over 70% months without fires) and the composite nature of fire mechanisms (should be an elusive non-linear function but we never know what elements contribute to this functions and what formation of this function should be). To tackle these issues, we have developed this hierarchy machine learning framework that aims to enhance the predictability of such extreme fire events, employing a range of non-linear functions and effectively addressing the scarcity of data with oversampling. Unlike previous applications of machine learning algorithms in wildfire science (Horton et al., 2021; Jain et al., 2020; Sayad et al., 2019; Yu et al., 2022), which often involve the direct use of a single algorithm or an ensemble of algorithms, our approach addresses the issue of data imbalance while simultaneously leveraging the physical processes of fire events. More specifically, our process initially predicts the likelihood of a fire occurrence under certain natural and anthropogenic conditions. We forecast the potential fire sizes only when a fire is

predicted to occur. More importantly, our framework incorporates a technique for quantifying uncertainty. This technique leverages the inherent ability to predict wrong classification likelihood (uncertainty) from Step One and then employs it as a broad-based adjustment to rectify error propagation in Step Two. It is a novel application in that we combine machine learning algorithms and uncertainty quantifications for the whole hierarchy system (framework) to improve the predictability of extreme BP fires.

Traditional machine learning algorithms, such as Random Forest (RF), Support Vector Machine (SVM), Artificial Neural Networks (ANN), Gradient boosting (GB), Convolutional Neural Networks (CNN), and Long-Short-Term Memory Networks (LSTM) and traditional data-driven time series forecasting algorithms such as Autoregressive Integrated Moving Average (ARIMA) and Seasonal ARIMA (SARIMA) can be applied in solving extreme event predictions, including BP fires. However, the choice of machine learning method largely depends on the nature of the data and the specific problem at hand. Here, we compared the directly employed various regression models on FireCCIV5.1 data to address the effectiveness of our framework. In the revised manuscript, we have revised the Methods part from **Lines 261- 267** and added a subsection of *4.3 validation* in the Results Section to address the framework's robustness and effectiveness.

For your second question, we appreciate your observation regarding the mention of smoldering fires in the Introduction section and its subsequent absence in our Method and Results sections. We assume that fires occurring within our research areas are predominantly smoldering, based on the defined characteristics of the peat soil (the proportion of histosol soil). The boreal peatland areas we refer to in our manuscript are those containing over 30% histosol soil. Histosols are commonly found in boreal regions where the cool and damp conditions slow down the decomposition of plant material, causing an accumulation of organic matter. They typically form in environments such as bogs and fens, where a high-level water table and an abundance of sphagnum moss and other vegetation contribute to peat formation. The survival and growth of trees can be challenging in those environmental conditions due to factors such as high acidity, a lack of essential nutrients, and waterlogged environments, though certain adaptive species such as black spruce can thrive. Our manuscript limits the peatland area to regions with more than 30% histosol content, aiming to ensure the presence of adequate soil fuel for smoldering while limiting the aboveground fuel such as forests or grasses. As a result, we propose that BP fires in our designated areas are predominantly smoldering. While the possibility of aboveground fires cannot be ruled out, we currently lack sufficient data and access for verification, an area we intend to improve in the future. Thus, we mainly introduced the ML framework and datasets in the Methods section, addressed the framework's effectiveness in predicting extreme fires, and connected the screened key factors to possible pathways that may induce smoldering fires with existing studies. In the revised manuscript, we revised the Methods section and improved the Results section with more connections to smoldering fires. Those modifications are in **Lines 134 – 144.**

Regarding your third question on the ability of the Global Fire Emissions Database (GFED) or FireCCI products to distinguish between smoldering or flaming peatland fires, as of my knowledge, both these products primarily quantify the total fire activity and emissions. While these two products can provide extensive information about fire locations, extents, and sometimes even fire severity, distinguishing between different types of fires (such as smoldering vs flaming fires) is challenging, especially at the spatial resolutions typically

provided by these two projects. Peatland smoldering fires present a unique detection and characterization problem due to their subsurface burning nature. In theory, thermal sensors on satellites, such as those on MODIS, could discern heat sources by capturing the infrared radiation emitted by fires. However, identifying smoldering fires can be particularly difficult until the fire spreads from beneath the surface to the up surface. Smoke detection is an alternative way to distinguish smoldering fires, as smoldering fires tend to produce more smoke particles than flaming fires, especially carbon monoxide (CO). Thus, CO detectors could be more effective. However, it's important to note that the use of satellites for carbon detection is still in its early stages. In some cases, airborne and ground-based LIDAR system are being deployed in filed investigations to detect smoldering fires at landscape or regional scale, providing an additional layer of detection capability. In our revised paper, we addressed the current capabilities and limitations of these two products at the Data section **(Line 142 - 144)**

2. In Step One (the classification step), author presented the datasets and algorithm employed in the classification process, as well as the preprocessing methods utilized for input data. However, author did not provide sufficient information regarding how the prediction of fire occurrence was made. Please provide additional details regarding the methodology employed to predict fire occurrences.

**Response:**

In our study, we implemented a two-step hierarchy ML framework to predict both fire occurrences and fire sizes. The likelihood of 'occurrence' is determined by ML classification algorithms that are employed in Step One. Each of these algorithms utilizes distinct methods to calculate the probability of each data sample belonging to a specific class. Leveraging this probability, we ultimately derive binary classifications, which serve as our fire occurrence predictions, indicating the presence or absence of fires each month at every geographical location.

To predict the likelihood of fire occurrence, we employed a variety of factors, such as weather conditions (temperature, humidity, wind speed), vegetation, human activities, and soil moisture. Previous research has identified these variables as possible contributors to fire occurrence. The classification models were trained with a random subset of the historical data, where the aforementioned factors serve as the independent variables and the occurrence of fire is the dependent variable. Here, the absence and presence of fire occurrence is defined by 0 and 1, respectively, according to the observational satellite-detected fires (e.g., GFED burned area size, C emission size, or MODIS burned date). If the Carbon emissions/Burned area value is greater than 0 or the data record is predicted as class 1, this indicates the presence of fire occurrences. Conversely, if the Carbon emissions/Burned area value is 0 or is predicted as class 0, it signifies a fire-free month or a month without any fire occurrence. To overcome the overfitting problem induced by the imbalanced data of fire presence and absence (over 70%), we utilized an oversampling algorithm to boost the data amount with fire occurrence. The Random Forest model, which has performed the best for its robustness and accuracy (over 0.9) among six fundamentally different algorithms, was ultimately selected for fire occurrence. At the training stage, the model learns the patterns and relationships between the environmental variables and fire occurrence from the historical data. As we don't get future climate and environment data, we didn't include the fire occurrence forecast for the future.

We have modified the Methods Sections, and the corresponding modifications can be found at **Lines 160 – 184 and Lines198 – 208.**

3. Line 195- "Fire size that might be caused by the wrong classification (namely, no fire happens in reality) could be expressed by EPnm". Since this analysis takes place during the prediction period, it may not be possible to determine which input variables (Xfm or Xnm) were wrongly classified if there is no real-time information on whether a fire has occurred or not. Please provide more explanation. And the meaning of True Positive (TP) in Equation 9 is not clear, it is uncertain how the authors obtained the "true" data during the prediction period.

**Response:**

In Line 195 (**the revised version of Line 220-221**), we introduced the concept of *error prediction in months-with-fire* ( $EP_{mn}$ ), referring to the instances where our model inaccurately predicted a fire at months when none fires occurred. Indeed, for every month being predicted (either $X_{mf}$ or $X_{mn}$) at Step One, due to our reliance on classification results as the 'true predictions' for evaluating fire sizes at Step Two, each month carries a possibility of being misclassified. More specifically, the term "wrong classification" associated with $EP_{mn}$, denotes that the model predicted a fire event when none actually occurred, while the "wrong classification" in the context of $EP_{mf}$ denotes a situation where the model failed to predict a fire event that did, in fact, occur. These scenarios underpin the basic assumption for our error correction process.

You are correct in highlighting that without real-time fire occurrence data. Due to the limitation of future data, we just utilized the diagnostic data, and the fires predicted at the testing stage is also based on historical data. In our study, however, we mitigated the limitation of prognostic data by using a holdout dataset (i.e., the testing dataset) to validate the performance of our model. This holdout dataset contains historically recorded fire events that were not used in the training phase. We identify the 'wrong classification' likelihood through the confusion matrix at the classification step, and then applied the uncertainty evaluation matrices to the regression step to correct the error propagation. I think, if there is a chance to apply future forecasting under this framework, a dynamic simulation can be helpful, in which case we will apply current uncertainty evaluation matrices as an initial value and then continuous adjust it with the input of future data (or real-time data).

Regarding your question about Equation 9 and True Positives (TP) comes from the confusion matrix for a binary classifier (with positive and negative classes), which can be visualized as follows:

|                      | Predicted Positive  | Predicted Negative  |
| -------------------- | ------------------- | ------------------- |
| Actual Positive (P)  | True Positive (TP)  | False Negative (FN) |
| Actual Negative (N)  | False Positive (FP) | True Negative (TN)  |

The TP denotes instances where the model correctly predicted a fire, and a fire did occur actually. As with the method described for $EP_{mn}$ and $EP_{mf}$, we identified these two items by

comparing the model's predictions against our holdout dataset at the testing stage (namely the predictive stage). This predictive stage also relies on historical data, as we aim to evaluate the ML model performances. However, we never use a further data or real-time data for future forecasting. Therefore, the "true" data in our case are historically recorded fire events from satellite-based products, and the model's 'predictions' were evaluated against this information.

4.  If you do not consider the fire occurrence from Step One, just predicted fire emissions or burned area based on training datasets listed in Table S1, is there a significant difference between the predicted results and the actual fire occurrences, which resulted in this approach?

    **Response:**

    Yes, the difference is significant, if we directly predict fire sizes using the datasets listed in Table S1 in regression models rather than processing from classifications.

    Here, we have tested 15 regression models, including the simple linear (LinR), ridge, least absolute shrinkage and selection operator (LASSO), adaptive boosting (Ada), gradient boosting (GBR), Bagging(Bag), Random Forest(RF), Bayesian regression (Bayes), Elastic Net (EN), Kernel Ridge (Kernel), Decision tree (DT), CatBoost (CBR), light gradient boosting (LGBR), and extreme gradient boosting (XGBR), to examine the model performance of direct application, not using the hierarchy framework as we did. Evaluation matrices, the mean squared error (MSE), mean absolute error (MAE), and the explained variance score–the R-square ($R^2$) are shown in Table 1.

    We could see that most models have bad performance–very large bias and very small value of explained variances–at both the training and testing stages. If utilizing our ML framework, the explained variances can generally be over 50%, at the testing stage, and, for this case, it is around 1%, much smaller than those using our framework.

    Exceptional examples, in this experiment, are that the RF, BGA, CBR, LGBR, and Stack models have relatively good performances in the training stage but very bad performances in the testing stage, which are typical overfitting problems due to severe imbalance of data, indicating very bad predictability of these models. Consequently, we couldn't apply those models directly due to bad behaviors. Yes, you got that right, and this is the reason why we designed a hierarchy framework to predict BP fires, especially in terms of their sizes, rather than applying the models directly.

    We updated our manuscript in **Lines 263 – 268 and Lines 335 – 345**.

Table 1 Evaluation matrices of regression models with direct applications.

| Model | Stage | MSE | MAE | $R^2$ |
|-------|-------|-----|-----|-----|
| Ada | training | 953.03 | 6.84 | 0.39 |
| | testing | 1068.78 | 6.91 | 0.12 |
| Bag | training | 154.47 | 2.38 | 0.90 |
| | testing | 927.57 | 6.31 | 0.23 |
| Bayes | training | 1450.62 | 11.26 | 0.07 |
| | testing | 1142.95 | 10.85 | 0.05 |
| CBR | training | 63.89 | 3.13 | 0.96 |
| | testing | 810.75 | 6.49 | 0.33 |
| DT | training | 0.00 | 0.00 | 1.00 |
| | testing | 1841.74 | 7.53 | -0.62 |
| EN | training | 1498.48 | 9.52 | 0.04 |
| | testing | 1159.71 | 8.99 | 0.04 |
| GBR | training | 1302.99 | 8.55 | 0.17 |
| | testing | 1123.06 | 8.31 | 0.07 |
| Kernel | training | 1450.86 | 11.41 | 0.07 |
| | testing | 1147.22 | 11.03 | 0.05 |
| Lasso | training | 1452.72 | 11.06 | 0.07 |
| | testing | 1141.87 | 10.65 | 0.06 |
| LGBR | training | 344.76 | 4.51 | 0.78 |
| | testing | 761.45 | 7.03 | 0.37 |
| LinR | training | 1448.34 | 11.45 | 0.07 |
| | testing | 1146.44 | 11.04 | 0.05 |
| RF | training | 139.32 | 2.40 | 0.91 |
| | testing | 928.29 | 6.42 | 0.23 |
| Ridge | training | 1450.14 | 11.38 | 0.07 |
| | testing | 1144.92 | 11.00 | 0.05 |
| Stack | training | 244.76 | 2.57 | 0.84 |
| | testing | 804.62 | 5.55 | 0.33 |
| XGBR | training | 1550.82 | 6.73 | 0.01 |
| | testing | 1198.16 | 6.25 | 0.01 |

5. Line 136- "The evaluation metrics from Step One, denoting the model uncerainties, are used at Step Two to correct fire size prediction uncertainties. The two-step ML framework is detailed in Figure 1". Please provide more explanation about what the meaning of the model uncertainties in Step One is.

**Response:**

We apologize for the unclear statement. Here are some supplementary explanations.

In the context of our two-step hierarchy Machine Learning (ML) framework, the 'model uncertainties' actually mean the error propagation from Step One, referring to the inherent inaccuracies or errors in predicting fire occurrences using ML classification models. Theoretically, there are no perfect predictive classification models, although the best-performed RF and BAG have very high accuracy in our case and they still have a small likelihood of introducing commission error and omission error, which is described in the term 'uncertainty' in this study. These uncertainties arise due to a variety of factors, such as limitations in the data, simplifications in the model, and inherent randomness of the BP fire events caused by changing climate or other nature factors.

More specifically, the classification accuracy from the confusion matrix in Step One is described below:

False Positives (FP): the number of cases where the model predicted a fire occurrence, but no fire actually occurred.

False Negatives (FN): the number of cases where the model did not predict a fire occurrence, but a fire actually did occur.

True Positive (TP): the number of cases where the model truly predicted a fire occurrence.

True Negative (TN): the number of cases where the model didn't predict a fire that is aligned with reality.

As our best-performed and most robust classification accuracy exceeds 90%, in Step Two, we assumed that each classification is hundred percent true: for each data record, if the classification prediction is 0, then we assumed that there will be no fire happen and the fire size is 0; conversely, if the classification prediction is 1, then we assumed that there will be fire and we utilized regression models to predict how large the fire size is. But in fact, our assumption is flawed, and we didn't include the uncertainty from wrong classifications. Thus, we correct the regression predictions with classification uncertainties. For all the data records with the predicted classification of 1, there is a probability of PPV being classified as true and there is a probability of FDR to be classified as wrong, where PPV is the Positive Predictive Value:

PPV $= \frac{TP}{TP+FP}$; and FDR is the False Discovery Rate: FDR $= \frac{FP}{TP+FP}$. For all the data records with predicted classification of 0, there is a probability of NPV to be classified as true and a probability of FOR to be classified as wrong, where NPV is the Negative Predictive value: NPV $= \frac{TN}{FN+TN}$, and the FOR is the False omission rate: FOR $= \frac{FN}{FN+TN}$ . Thus, the adjusted fire size predictions from the regression models are presented as :

$$AP'^{r}_{mp} = \begin{cases} PPV \times P^{r}_{mf} + FDR \times EP^{r}_{mn}, & If\ Z'_{mp} = 1 \\ NPV \times P^{r}_{mn} + FOR \times EP^{r}_{mf}, & If\ Z'_{mp} = 0 \end{cases}$$

Where $Z'_{mp}$ stands for the classification predictions; $P^r_{mf}$ is the original predicted fire sizes for each record for months with fires; $P^r_{mn}$ is the original predicted fire sizes for each record for months without fires; $EP^r_{mn}$ and $EP^r_{mf}$ are uncertainties associated with wrong classifications.

We have updated the Methods Section, and the introduction to the ML framework, and the error corrections are updated in **Lines 145–239.**

6. Line 140- "(i.e., there are more nonoccurrence records than occurrence records)". This sentence need clarification.

**Response:**

Thank you for pointing out the need for clarification. Here, we aimed to indicate that there are over 70% of data records without fires while only 30% of data records are related to fires presences. In the revised manuscript, we added the description at **Lines 149 – 150**.

7. Line 145- Figure 1. Please increase the size of the text for better readability (also for Figures in SI). The resolution of the images is quite low.

**Response:**

Thank you for your valuable feedback. We sincerely apologize for any difficulty you've experienced in reading the text in Figure 1 and other supplementary figures due to their small size and low resolution. **We have updated them correspondingly in the revised MS, as well as below:**

[Figure]

8.  Line 149- "An oversampling algorithm called Synthetic Minority Oversampling Techniques (SMOTE) was applied onto the training dataset to address the imbalance between the two fire occurrence classes". Please elaborate on how an imbalance between different datasets can affect the model's performance in predicting fire occurrences and provide more details regarding procedures of using the SMOTE algorithm to address the imbalance problem.

**Response:**

Thank you for your question. The imbalanced dataset usually reflects a skewed data distribution or an uneven distribution of classes which often occurs in a classification problem. In our case, the fire occurrence presence and absence are a binary classification problem, in which we defined the fire absence and presence as class 0 and class 1, respectively. One of the biggest disadvantages that the imbalanced data may bring is an overfitting problem, in which the model has very high fitting accuracy at the training stage but very low predictive accuracy at the testing stage, indicating that the model doesn't have good predictability on new datasets.

In our case, utilizing the GFED BA data as a predictive variable and taking the training dataset without oversampling as an example, there are only 1656 data records with fires while the other 6448 records are months without fire; similarly, in the testing step, 710 and 2764 records are months of fire presences and fire absences. Basically, only 20% of months have fire occurrence presences. One limitation is that model fitting many more examples of the majority class (fire occurrence absence) during the training phase may cause it to be less effective at predicting the minority class (fire occurrence presence), resulting in the overfitting problem. Along with the overfitting problem, the model has a very high accuracy in the training stage but a very low accuracy at the testing/predicting stage, indicating a low predictability of the trained model on new datasets.

The Synthetic Minority Over-Sampling Technique (SMOTE) is a popular method for addressing this imbalance. The SMOTE algorithm works by generating synthetic samples from the minority class (fire occurrence presence in our case) rather than creating copies. The algorithm selects examples that are close to the feature space, drawing a line between the examples in the feature space and generating a new sample at a point along that line. Essentially, it creates "synthetic" fire occurrences that are plausible based on the existing fire occurrences. By oversampling the minority class in this way, the imbalance between the fire and non-fire occurrences in the training dataset is reduced.

Our application of SMOTE involves the following steps:

- *Identify the imbalance in our training dataset between fire occurrence presence and absence records.*
- *Apply the SMOTE algorithm to the fire occurrence presence records in our training dataset. The algorithm begins by choosing a minority class instance. Then, it identifies its 'k' nearest neighbors (we typically choose five for our model).*
- *Among these 'k' neighbors, one is selected randomly, and synthetic examples are created between the chosen instance and its neighbors.*
- *This process continues until the dataset is balanced, meaning there is roughly an equal number of fire occurrence presence and absence records.*

In our revised manuscript, we include additional details about the impact of dataset imbalance on model performance at Lines **335-343** and further clarify how we utilized the SMOTE algorithm to handle the data imbalance and overfitting at Lines **163-168**.

9. Line 217- "…could be one primary source of feature collinearity," This sentence need clarification.

**Response:**

Thank you for pointing out the need for clarification regarding this sentence. I can certainly provide more context, although the logic is not straightforward.

When we refer to the "Others group", we mean a collection of features that don't fall neatly into the other four feature categories: temperature, precipitation, air-dryness, and soil moisture. These features within the "Others group" include factors like wind speed, vegetation type, topographical features, etc. The "Others group" has a diverse set of features that each carry different physical meanings. For instance, wind speed might influence the spread of fires, while vegetation type can affect the likelihood of a fire starting in the first place. Because of this diversity, including the 'Others group' features will increase the challenge to manage collinearity not only within the group but also between groups.

In this research, the features in the other four groups are grouped by similar physical meanings, we can't directly apply traditional matrices (such as correlation matrix, variance inflation factor, etc.) to check collinearity among groups, thus conversely, we keep these features to check the grouped factor ranking consistency to examine the collinearity conversely, which is like reverse verification. Thus, despite the potential collinearity issue, we chose to keep the "Others group" features in our simulations due to their potential importance in predicting fire occurrences and consistent feature ranking.

In our revised paper, we have included some explanations from **Lines 249 – 252.**

10. What is the meaning of "training and testing" in Section 3 and 6 of SI, please provide clarification on this matter and explain the purpose of presenting both training and testing results in the figures. Also, what is the differences between "(a-1) observed" and "(a-2) observed" from Figures S3 to S12?

**Response:**

Thank you for your questions. I'll clarify each point.

The 'training and testing' terms mean the validated model accuracy on datasets from both the training and testing stages. Now, we have removed all the results related to 'training and testing', and the SI is updated.

**Our purposes were below:**

In the SI, we validated the model performance on different datasets (or stages), primarily on testing-dataset-only and all (including training and testing). In the supplementary

information, the Section S3, S6, and S10 presented the validation results of fire counts spatial distribution, the fire counts statistical distribution (namely the seasonality), and fire impacts statistical, respectively, *based on all datasets* (namely both training and testing datasets). Section S4, S5, and S8, presented the validation results of fire counts spatial distribution, the fire counts seasonality, and the fire impact (sizes, namely BA or C emissions) seasonality, respectively, *based on the testing dataset only*.

*Presenting both the training and testing results in the figures serves to show the model's performance at both stages, compared to the testing stage only.* This is important because while a model may perform well on the training data, it's the performance on the testing data that truly gauges its effectiveness. Good performance in the training phase but poor performance in the testing phase might suggest overfitting, where the model has become too complex and is fitting the noise in the training data rather than the underlying pattern. Thus, knowing the model performance with goodness and badness, by comparing the result differences on both all (training and testing) and testing-only that induced by training, can show valuable information to illustrate model performance.

**Regarding your question about Figures S3 to S12,** "(a-1) observed" and "(a-2) observed" represent two different observation datasets used in the study. In Figures S3 and S12, he first and third columns are regions in North America (including the Hudson Bay area and Canada) while the second and fourth columns are areas in Western Siberian. Thus, "(a-1) observed" and "(a-2) observed" means the observations in North America and Western Siberian, respectively.

**In the revised SI, we have modified the subtitles which are marked in Red, and we have added some descriptions at the beginning of each section.**

11. Figure S35 lacks axis titles.

    **Response:**

    We apologize for the oversight. The absence of axis titles in Figure S35 (**In revised version, Figure S22**) are updated as below. we greatly appreciate your attention to detail in pointing this out.

[Figure]

[Figure]

[Figure]

Thank you very much for providing valuable observations, suggestions, and comments. Your feedback is instrumental in improving the clarity and quality of our work, and we are thankful for your contribution.

**References**

Horton, A.J., Virkki, V., Lounela, A., Miettinen, J., Alibakhshi, S., Kummu, M., 2021. Identifying Key Drivers of Peatland Fires Across Kalimantan's Ex-Mega Rice Project Using Machine Learning. Earth and Space Science 8, e2021EA001873. https://doi.org/10.1029/2021EA001873

Jain, P., Coogan, S.C.P., Subramanian, S.G., Crowley, M., Taylor, S.W., Flannigan, M.D., 2020. A review of machine learning applications in wildfire science and management. Environmental Reviews. https://doi.org/10.1139/er-2020-0019

Sayad, Y.O., Mousannif, H., Al Moatassime, H., 2019. Predictive modeling of wildfires: A new dataset and machine learning approach. Fire Safety Journal 104, 130–146. https://doi.org/10.1016/j.firesaf.2019.01.006

Yu, Y., Mao, J., Wullschleger, S.D., Chen, A., Shi, X., Wang, Y., Hoffman, F.M., Zhang, Y., Pierce, E., 2022. Machine learning–based observation-constrained projections reveal elevated global socioeconomic risks from wildfire. Nat Commun 13, 1250. https://doi.org/10.1038/s41467-022-28853-0

**Reviewer 2:**

The authors presented a two-step machine learning framework to quantify wildfire drivers and predictability in boreal peatlands: The first step use classification models to identify fire occurrences, and the second step use regression models to the burned area and C emissions, as well as the relative importance of environmental drivers. This work tested multiple datasets and tested various ML methods. While the effort is appreciated, it lacks a clear rationale for the choice of methods and what is novel compared to previous studies.

In its current form, the manuscript is a bit convolved and hard to digest (too much jargon and acronyms). There are redundancies in many places. The results could be further distilled to provide more insights. Analysis on a regional scale would be interesting. Do the models perform better in some regions than others? Currently, data are randomly split for training (70%) and predicting (30%); if we use recent years with mega boreal fires as predicting, how well would the models perform?

Smoldering is a key focus of the introduction and is mentioned as a challenge in the discussion. However, it is not clear how the results are relevant to it. C emissions are mentioned, but results are not shown.

**Responses to reviewer#2:**

We appreciate your thoughtful and comprehensive feedback on our manuscript. Your suggestions provide valuable direction for improvement.

**Regarding the choice of methods,** we agree that we need to provide a clearer rationale. One reason we constructed the two-step hierarchy machine learning framework is that the BP peatland fires differ from fires in fire-prone areas (such as Africa Savannah and the western United States), which are usually extreme fire events. Taking the satellited-originated fire product–the GFED4.1 burned area–as an example, Basically, only 20% of months have fire occurrence presences, and in the training stage, there are only 1656 data records with fires while the other 6448 records are months without fire; similarly, in the testing step, 710 and 2764 records are months of fire presences and fire absences. One limitation of this skewed data distribution is that model fitting many more examples of the majority class (fire occurrence absence) during the training phase may cause it to be less effective at predicting the minority class (fire occurrences presence), resulting in the overfitting problem. Along with the overfitting problem, the model has a very high accuracy in the training stage but a very low accuracy at the testing/predicting stage, indicating a low predictability of the trained model on new datasets. If we directly apply machine learning algorithms to predict fire sizes, not using our two-step framework, the predictability of models would be very low, and overfitting is a major problem needed to be solved. For this sake, we have tested 15 regression models, including the simple linear (LinR), ridge, least absolute shrinkage and selection operator (LASSO), adaptive boosting (Ada), gradient boosting (GBR), Bagging(Bag), Random Forest(RF), Bayesian regression (Bayes), Elastic Net (EN), Kernel Ridge (Kernel), Decision tree (DT), CatBoost (CBR), light gradient boosting (LGBR), and extreme gradient boosting (XGBR), to examine the model performance of direct application, not using the hierarchy framework as we did. Evaluation matrices, the mean squared error (MSE), mean absolute error (MAE), and the explained variance score–the R-square (R2) are shown in Table 1.

Table 1 Evaluation matrices of regression models with direct applications.

| Model | Stage | MSE | MAE | R^2 |
|-------|-------|-----|-----|-----|
| Ada | training | 953.03 | 6.84 | 0.39 |
|  | testing | 1068.78 | 6.91 | 0.12 |
| Bag | training | 154.47 | 2.38 | 0.90 |
|  | testing | 927.57 | 6.31 | 0.23 |
| Bayes | training | 1450.62 | 11.26 | 0.07 |
|  | testing | 1142.95 | 10.85 | 0.05 |
| CBR | training | 63.89 | 3.13 | 0.96 |
|  | testing | 810.75 | 6.49 | 0.33 |
| DT | training | 0.00 | 0.00 | 1.00 |
|  | testing | 1841.74 | 7.53 | -0.62 |
| EN | training | 1498.48 | 9.52 | 0.04 |
|  | testing | 1159.71 | 8.99 | 0.04 |
| GBR | training | 1302.99 | 8.55 | 0.17 |
|  | testing | 1123.06 | 8.31 | 0.07 |
| Kernel | training | 1450.86 | 11.41 | 0.07 |
|  | testing | 1147.22 | 11.03 | 0.05 |
| Lasso | training | 1452.72 | 11.06 | 0.07 |
|  | testing | 1141.87 | 10.65 | 0.06 |
| LGBR | training | 344.76 | 4.51 | 0.78 |
|  | testing | 761.45 | 7.03 | 0.37 |
| LinR | training | 1448.34 | 11.45 | 0.07 |
|  | testing | 1146.44 | 11.04 | 0.05 |
| RF | training | 139.32 | 2.40 | 0.91 |
|  | testing | 928.29 | 6.42 | 0.23 |
| Ridge | training | 1450.14 | 11.38 | 0.07 |
|  | testing | 1144.92 | 11.00 | 0.05 |
| Stack | training | 244.76 | 2.57 | 0.84 |
|  | testing | 804.62 | 5.55 | 0.33 |
| XGBR | training | 1550.82 | 6.73 | 0.01 |
|  | testing | 1198.16 | 6.25 | 0.01 |

We could see that most models have bad performance–very large bias and very small value of explained variances–at both the training and testing stages. Exceptional examples are that the RF, BGA, CBR, LGBR, and Stack models have relatively good performances in the training stage but very bad performances in the testing stage, which are typical overfitting problems due to severe imbalance of data, indicating very bad predictability of these models. Consequently, we couldn't apply those models directly due to bad behaviors. This is the reason why we designed a hierarchy framework to predict BP fires, especially in terms of their sizes, rather than applying the models directly. **The novelty of our study lies in the integration of these two steps, along with correcting the error propagations to combat the overfitting problem in extreme fire predictions.** While individual elements of our approach might have been used in previous studies, we believe our combined approach provides a comprehensive view of fire prediction in boreal peatlands.

In the revised manuscript, we clarify this problem in the manuscript in S4.3 Validation ( Lines 336–345).

Thanks for your suggestions on tailoring the manuscript. We reorganized the language and summarized the jargon and acronyms in a list of abbreviations in **Table S2**, trying to make the manuscript more accessible to a broader audience. **Additionally, the Methods Section has been reorganized and updated.**

Your suggestion to perform regional-scale analyses is insightful. If we had access to more detailed field measurements and BP fire records, it would be a fruitful avenue to explore. Our current research relies on datasets, especially the BP fire datasets, which are derived from satellite imagery. Taking into account the precision, consistency, and potential error margins of these datasets, we opted to focus on a broader regional scale. That said, we aspire to delve into more localized studies in the future to gain deeper insights into the mechanics of BP fires.

To address the question of regional modeling discrepancies, the data suggests that the fire regime in the Hudson Bay area is somewhat different from its counterpart in western Siberia. The latter witnesses a higher frequency of BP fire events, exhibits greater spatial heterogeneity, and has a higher frequency of extreme fires. Surprisingly, our model's spatial predictions remain largely robust and accurate across these regions, as depicted in the figures in Sections 3 and Section 6 in the revised SI. One imperfection is the model's overestimation of C emissions in parts of the Hudson Bay area.

For the data splitting questions, we did split the dataset randomly, allocating 70% for training and 30% for testing. Meanwhile, we also implemented a stratified approach for the target variable, ensuring the model trains with a comprehensive range of data features.

Thank you for your suggestion on further validation for large BP fires in recent years. Due to data availability constraints, using the same inputs wasn't feasible. To accommodate this and extend the data series, we replaced the GIMMS-3g NDVI and GPP with MOD13C2 NDVI and VODCA2 GPP data, respectively, as detailed in Wild et al., (2022). Although the VODCA2GPP has a longer time span, it doesn't offer global coverage, and data in most of our research regions are missing, thus we did a very coarse interpolation to gap-filling the data. Additionally, we interpolated the population density data to extend its timeline from 2016 to 2019. We have preprocessed the inputs, excluding records with the NAN data, and standardizing the features. Using our pre-established model, we then predicted fire occurrences for this period. The accuracy of these predictions is detailed below:

| | accuracy | recall | precision | f1score | roc_auc | kappa |
|---|---|---|---|---|---|---|
| **RF** | 0.470665 | 0.704082 | 0.119308 | 0.204041 | 0.574928 | 0.046986 |
| **BAG** | 0.480498 | 0.707483 | 0.121851 | 0.207896 | 0.581888 | 0.052045 |
| **KNN** | 0.566044 | 0.585034 | 0.125182 | 0.206235 | 0.574526 | 0.056441 |
| **LogReg** | 0.724025 | 0.343537 | 0.134667 | 0.193487 | 0.554068 | 0.06388 |
| **SVM** | 0.718781 | 0.37415 | 0.140306 | 0.204082 | 0.564841 | 0.074338 |
| **GNB** | 0.572599 | 0.5 | 0.11273 | 0.18398 | 0.54017 | 0.031699 |

From the fire occurrence prediction accuracy, we could see that our framework handled the overfitting problem well, even though it didn't perform as well as our previous experiments. The reasons could be two major reasons: one is the total new data set with new data features (like Descals et al., 2022 's paper showing that arctic fires increase incredibly after 2016,); another reason could be the very bad quality of GPP and population density data.

Regarding the concern about the smoldering-related text, we apologize for the inadequate assessment of our manuscript. One of our intentions behind developing this machine learning framework was to complement existing tools in identifying potential drivers of smoldering fires, especially given the uncertainties surrounding their mechanisms in the context of global change. We've incorporated a

great number of factors believed to influence smoldering fires into our model. By identifying the most influential among these, we then sought to align them with existing scientific theories to understand their potential roles in driving smoldering fires. Certainly, the mechanics of smoldering fires demand a more in-depth exploration or quantification of the causalities, such as employing Structural Equation Modeling to quantify the pathways influenced by the identified factors. While this analysis falls beyond the scope of our current machine-learning endeavor, we are interested to explore this topic as a standalone project. If our data permits, we're also considering a more site-specific analysis to provide a richer discussion and quantification on the influence of the identified factors in triggering smoldering fires in future revisions. We added some relative discussions in **Lines 407 – 416 and Lines 420 – 425.**

Thank you for highlighting the omission of carbon emissions in our primary discussion. There are two principal reasons for this decision. Firstly, carbon emissions show a strong correlation with the burned area, which may make the presentation redundant in some respects. Secondly, when considering smoldering fires, factors intrinsic to peatlands such as peat depth and carbon density are pivotal in determining carbon emissions. However, current emission products do not account for these crucial variables. As a result, the uncertainty associated with carbon emission data tends to be significantly greater than that of the burned area product. Given these considerations, we opted to focus on the burned area in the main text and have included carbon emission results in the supplementary material for those interested in delving deeper.

Once again, we appreciate your constructive feedback, which is helpful in enhancing the quality and impact of our study. we will address other questions below:

1.  -Line 164-166: I would start this paragraph starting from "In Step Two…"

    **Response:**

    Thank you for your suggestion. In the revised manuscript, we have reorganized and updated the Methods Section.

2.  -Line 173-200: in my opinion, this part is unnecessarily hard to follow. The essence of all these could be summarized for better readability

    **Response:**

    Thank you for your feedback. We understand the importance of clarity and readability, especially in a lengthy technical description. We have added a summarized introduction to each subsection in the Methods.

3.  -Line 335-341: reiterating introduction. The discussion is a bit disconnected from the work done here.

    **Response:**

    We appreciate your feedback on this section. We have removed this paragraph and started the discussion with our findings directly in 5.1., beginning from **Line 378 in the revised manuscript**.

Regarding for the disconnections, we have added some discussions and limitation on the causality problem and data limitations, which is presented in **Lines 420 – 425.**

4. Figure 1: hard to read

**Response:**

Thank you for your feedback. We have updated the Figure 1 as below.

[Figure]

5. Figure 2: are there regional differences in their behavior?

**Response:**

The data indicates that the fire regime in the Hudson Bay area slightly differs from that in western Siberian. The West Siberian has more BP fire events (higher frequency), larger spatial heterogeneity, and more large extreme fire sizes.

Surprisingly, our model didn't show significant performance discrepancies in spatial predictions (Figures in Section 3 and Section 6 in the revised SI), and model accuracy are overall good in most areas. The only exception is the less-accurate predictions (mainly caused by overestimation) of C emissions in part of the Hudson Bay area.

6. Figure 3: it is not a very informative plot, as they all look similar.

   **Response:**

   Thanks for your insights. Yes, the regression models have very similar performance. The reason could be that the input data for each regression model are re-subtracted according to the classifications. Only the data records with predicted fire presence will be included in the regression model, which largely refined the input data. In addition, for all the regression models, we utilized random data splitting with a stratified target technique, which keeps the data features, between the training and testing data, are very similar and facilitates the regression model performances.

7. Figure 4: maybe focus on the best-performing models? Or use different regional subsets of samples to see if there are differences?

   **Response:**

   Thank you for your suggestions on improving Figure 4. For the best-performed model– Random Forest, the feature importance is ranked as below:

[Figure]

   The VPD has a dominant role with the largest importance, while other factors share similar importance.

8. Figure 5: it is not clear how the results of this paper are tied to all the processes shown here (only a tiny fraction of the variables are tested)

   **Response:**

   Thank you for your thoughtful critique of Figure 5. In Figure 5, we mainly discussed how can the model identify key factors (marked in blue) that participate in pathways inducing smoldering fires, by connecting existing research and theories.

Due to limited data availability, it is challenging to validate each pathway. But one highlight point is that we indicated the potential impacts of seasonal freeze-thaw cycles, which are rarely studied in smoldering fires. To quantify the causalities of our identified key factors to smoldering fire pathways, we will need more data availability and statistical investigation in the near future.

**Citation**: https://doi.org/10.5194/gmd-2023-14-RC2

Descals, A., Gaveau, D.L.A., Verger, A., Sheil, D., Naito, D., Peñuelas, J., 2022. Unprecedented fire activity above the Arctic Circle linked to rising temperatures. Science 378, 532–537. https://doi.org/10.1126/science.abn9768

Wild, B., Teubner, I., Moesinger, L., Zotta, R.-M., Forkel, M., van der Schalie, R., Sitch, S., Dorigo, W., 2022. VODCA2GPP – a new, global, long-term (1988–2020) gross primary production dataset from microwave remote sensing. Earth System Science Data 14, 1063–1085. https://doi.org/10.5194/essd-14-1063-2022

---

## Author Response (AR2)

*Reviewer #3:*

The manuscript submitted by Tang et al. presents a ML framework for improving the predictability of fire occurrences and sizes in boreal peatlands. Having gone through the first round of referee comments, I'm satisfied with the changes incorporated by the authors as well as the much improved discussion of the results.

However, I would like to highlight two minor areas of improvement for the manuscript before it is accepted for publication.

a) In the section on ML predictability (lines 379-380), while the authors have cited some papers as examples of ML application in fire modeling, they do not cite several recent papers that have developed interpretable ML models for global burned area prediction and stochastic fire prediction. These include but are not limited to:

https://gmd.copernicus.org/articles/16/869/2023/

https://gmd.copernicus.org/articles/16/3407/2023/

b) There should at least be some minimal documentation of how to navigate/run the analysis data and ML models provided in the GitHub repository -- there is none currently.

*Respond to reviewer #3:*

We are grateful for your constructive comments, which are very valuable in improving our manuscript. We have taken the following actions to enhance our manuscript:

a) Expanded References on ML Predictability (Lines 379-385):

We appreciate your recommendation of several key recent publications related to interpretable ML models for predicting global burned areas and stochastic fire occurrences. We have incorporated these references into our discussion to provide a more comprehensive overview of recent research and revised lines 379-384 (revisions are marked in blue) accordingly.

b) Enhanced GitHub Repository Documentation:

Your emphasis on the necessity of documentation has been well noted. In compliance with the Journal's policies and the Editor's requirements, we have archived the data, code, and some initial results on Zenodo, now including documentation (README.txt) in Version 2 (under Tangetal2023/Scripts/), which can be found at https://zenodo.org/record/7754018#.ZBi62uyZPK0. In addition, we have updated our GitHub repository. Due to constraints on data size, we have provided a README.txt file there only, which provides information for installation and running processes.

---

## Author Response (AR3)

1. In your manuscript, please use full first names for all authors. Although references are still based on initials, we will use full first names on the title page of your paper.

   **Response:** Thank you for your reminder. We confirm that full first names for all authors have been included on the title page of our manuscript.

2. Please ensure that the reproduction rights for all figures have already been secured and that maps and aerials include the required copyright statements or credits as requested by the providers.

   **Response:** We have ensured that reproduction rights for all figures are secured. Data source used in this study has been cited.

3. Before file upload, please consider submitting data sets, model code, or video supplements to reliable repositories, receive DOIs, and cite these assets in your manuscript including entries in the reference list.

   **Response:** Data sets and model code have been archived on Zenodo with assigned DOI, which has been included in the *Code and data availability* section.

4. To promote your work, please provide a 500-character short summary during production file upload and consider producing a short video abstract. Upload your video abstract to an appropriate video portal, provide the link/DOI during production file upload, and we will embed your video in your article's web page.

   **Response:** Below is the short summary.

   The study introduces TSECfire v1.0, a hierarchical error-correcting machine learning framework for predicting extreme boreal peatland fires. It emphasizes the dominant role of temperature and air dryness in BP fires, surpassing precipitation, wind speed, and human activities in inducing peatland fires. The study's unique approach lies in its two-step error-correcting framework, achieving over 80% accuracy in predicting rare and extreme fire occurrences and fire sizes. The paper also discusses two fire mechanisms that are with- and without frozen-thaw effects in understanding smoldering fires.